# Analysis of Fatty Acid Composition and Volatile Profile of Powder from Edible Crickets (*Acheta domesticus*) Reared on Apple By-Products

**DOI:** 10.3390/foods13111668

**Published:** 2024-05-26

**Authors:** Io Umebara, Keiko Akutsu, Misako Kubo, Akihiro Iijima, Ren Sakurai, Hirofumi Masutomi, Katsuyuki Ishihara

**Affiliations:** 1Research and Development Division, Calbee, Inc., 23-6 Kiyohara-Kogyodanchi, Utsunomiya 321-3231, Tochigi, Japan; i_umebara@calbee.co.jp (I.U.); k_ishihara@calbee.co.jp (K.I.); 2FUTURENAUT Co., Ltd., 1300 Kaminamienomachi, Takasaki 370-0801, Gunma, Japan; a-iijima@futurenaut.co.jp (A.I.); ren_sakurai@futurenaut.co.jp (R.S.)

**Keywords:** edible insect, edible insect flavor, apple by-products, fatty acids, volatile profile, gas chromatography, *Acheta domesticus*

## Abstract

Edible crickets have recently been used as a new alternative protein source with high nutritional value. The nutritional and flavor-related value of edible crickets varies greatly depending on the species, growth conditions and processing conditions. However, few studies have investigated the effects of the diet fed to crickets during their growth phase on flavor. Therefore, in this study, we characterized the flavor-related factors of powder from crickets reared on apple by-products (ACP) by comparing them with those of powder from crickets reared on a control diet (CCP). The fatty acid composition and volatile compounds of each powder were determined using gas chromatography and mass spectrometry, followed by sensory analysis and color measurement. A decrease in unsaturated fatty acids, specifically γ-linolenic acid, α-linolenic acid, arachidonic acid and docosahexaenoic acid, was observed in ACP. A total of 50 volatile compounds were identified, of which 11 were present in only ACP, while 39 were found in both powders. The sensory analysis showed that the overall balance score of ACP was higher than that of CCP, and according to the color measurements, ACP was darker than CCP. These differences between CCP and ACP might have been due to the differences in the chemical composition of the diets fed to the crickets during their growth phase. The results of this study suggest that one of the factors determining the food value of edible crickets, especially in terms of flavor, is the diet they are fed during their growth phase.

## 1. Introduction

The increase in the world population is predicted to cause an imbalance between protein supply and demand—the so-called protein crisis—and securing sufficient protein sources has become an urgent issue. However, producing the current major protein sources, such as beef and pork, requires significant amounts of feed, water, land and energy, which limits production [1]. Therefore, new protein sources are needed.

With the rapid development of plant-based and cultured meats as new protein sources in recent years, insect-based food is also attracting attention. Insects are rich in protein and can be processed into powder form easily, making them widely used as a nutrient fortifier in a variety of processed foods, such as pasta, protein bars and snack foods [2,3]. In terms of environmental impact, it is known that the energy required for their growth and the gas emissions produced during their development are much lower than those of conventional livestock [4,5]. In addition, food waste can be used in their diet during the growth phase, which may contribute to solving the food waste problem [6,7,8].

Crickets are known to be among the most protein-rich insects, containing approximately 50–70% protein in dry matter equivalent [9]. They also contain approximately 48–54% essential amino acids, approximately 1.3–3.1% minerals and approximately 11–24% fatty acids in dry matter equivalent, making them a promising new alternative protein source with high nutritional value [10,11].

Edible crickets are known to vary greatly in terms of nutrients depending on external factors, such as the diet they are fed during the growth phase, post-harvest processing conditions and storage conditions, so the quality of food products produced from the same cricket species may not always be the same [12,13,14,15,16,17]. The fatty acid composition, especially omega-3 fatty acid, fluctuates in crickets reared on diets containing small amounts of various oils, such as sunflower oil or fish oil [12,13]. Different temperatures selected for the drying of crickets have a significant effect on the protein content and fatty acid composition; a decrease in the level of various nutrients was observed at higher treatment temperatures [14]. In a study where crickets were reared on a diet in which fresh pumpkin pulp or dried pumpkin pulp powder partially replaced the general formula diet, the crickets’ nutritional protein, vitamin and mineral contents were significantly different from those of crickets reared exclusively on the control diet [15]. In addition, diets derived solely from food by-products have been shown to significantly alter the fatty acid composition in crickets [16].

Food value is not only based on nutrients but also on organoleptic properties such as appearance and flavor. In particular, flavor is one of the most important factors influencing consumer acceptance. Among known factors, the composition of volatile compounds such as terpenes, Maillard reaction products (MRPs) and secondary oxidation products (SOPs) greatly affects flavor. A comprehensive review focusing on volatile compounds in edible insects reported that the composition of volatile compounds is greatly affected by insect species or processing conditions, as MRPs and SOPs are generated due to temperature effects [17]. However, most studies focusing on the organoleptic properties of edible crickets have compared crickets in different rearing conditions, under various food processing conditions, etc., while few studies have focused on the effects of the external environment, especially the diet, during the growth phase on these properties.

Therefore, our aim in this study was to investigate the effects of different diets on the flavor of crickets. More specifically, powder from edible crickets reared on apple by-products, a type of food waste, was characterized in terms of fatty acid composition and volatile compounds to evaluate its food value, especially in terms of flavor. Sensory analysis and powder color measurement were also performed.

## 2. Materials and Methods

### 2.1. Preparation of Cricket Powder

Edible crickets (*Acheta domesticus*) and the control diet were obtained from FUTURENAUT Co., Ltd. (Gunma, Japan). Apple by-products, mainly composed of apple cores, were residual products generated during the production of dried apple snacks at Calbee, Inc. (Tokyo, Japan). The control crickets were raised for 4 weeks on a control diet consisting of corn, wheat, fish meal, soybean meal, rice bran wax and animal fat. The apple-fed crickets were raised on the control diet for 3 weeks and then on apple by-products for 1 week. After harvest, the insects were frozen (ICSD-14A-W; IRIS OHYAMA, Miyagi, Japan), sterilized in boiling water for 20 min, and then dried (OFWP-600V-R; AS ONE, Osaka, Japan) at 100 °C for 8 h. Then, the crickets were ground with a food processor (MK-K48P; Panasonic, Osaka, Japan) to obtain powder, which was kept in a −20 °C freezer (HF-120S3; Hoshizaki, Aichi, Japan) until analysis. Three samples from each of the different production lots were prepared for analysis. All reagents used in this study are listed in Appendix A.

### 2.2. Fatty Acids

The total lipids of the ground samples were extracted in accordance with Folch’s extraction procedure [18]. Briefly, 1 g of sample powder was homogenized for 3 min with 25 mL of chloroform/methanol (2:1 *v*/*v*). After centrifugation, the supernatant was collected. The extraction process was performed twice. The resulting solution was washed with 0.88% potassium chloride aqueous solution and dried over anhydrous sodium sulfate, and the solvent was evaporated.

Fatty acid methyl esters (FAMEs) of the total lipid extract were prepared using two-step transesterification in potassium hydroxide methanolic solution (KOH/MeOH) and sulfuric acid methanolic solution (H_2_SO_4_/MeOH) [19,20]. Briefly, 50 µL of total lipid extract was mixed with 500 µL of 0.4 M KOH/MeOH; the resulting mixture was vortex-mixed for 30 s and kept at room temperature for 10 min. Then, 550 µL of H_2_SO_4_/MeOH was added to the mixture, which was then incubated at 70 °C for 30 min. To collect the FAMEs, liquid–liquid extraction was carried out by adding 500 µL of *n*-hexane and vortex-mixing for 30 s. After centrifugation, the supernatant was collected. The extraction process was performed four times. The solvent was removed under a gentle stream of nitrogen. Methyl nonadecanoate (C19:0) was added to the extract as an internal standard and then dissolved with *n*-hexane.

The FAMEs were analyzed using a Thermo Scientific TRACE^TM^ GC Ultra gas chromatograph (Thermo Fisher Scientific, Waltham, MA, USA) equipped with TRACE^TM^ TR-FAME GC Columns (30 m × 0.25 mm × 0.25 µm; Thermo Fisher Scientific) and a flame ionization detector (FID) (Thermo Fisher Scientific, Waltham, MA, USA). Helium was used as the carrier gas at a flow rate of 1.0 mL/min. The oven temperature program consisted in increasing the temperature from 100 °C (0.2 min) to 180 °C at 1.0 °C/min and to 240 °C (15 min) at 10 °C/min. Sample introduction was accomplished under the following conditions: split/splitless injector type kept at 240 °C, split injection mode, 1:10 split ratio and 1 µL injection volume. The FAMEs were identified by comparing their retention time with that of known standards (Supelco 37 Component FAME Mix).

### 2.3. Volatile Compounds

The extraction of volatile compounds was performed using an Agilent 8697 headspace sampler integrated with the Agilent 8890 GCMS system (Agilent Technologies, Inc., Santa Clara, CA, USA). An aliquot of 2 g of each sample was introduced into a 20 mL headspace vial and incubated for 5 min at a temperature of 40 °C. An SPB^®^-1 SULFUR Capillary GC Column (30 m × 0.32 mm × 4.00 µm; Supelco Inc., Bellefonte, PA, USA) was used for the separation of the volatile compounds with helium as the carrier gas, in splitless mode, at a flow rate of 2 mL/min. The oven temperature program consisted in increasing the temperature from 45 °C (1.5 min) to 250 °C (10 min) at 12 °C/min. The MS was operated in single-quadrupole mode in EI (electron impact) mode at 70 eV. The ion source and transfer line temperatures were 280 and 250 °C, respectively. The analyte identity was confirmed by matching the EI spectrum with reference spectra in the NIST27 mass spectra libraries. The relative area (%) for volatile compounds was calculated based on the total area of the compound peaks. Compounds identified in at least three of the five total ion chromatograms were used in the calculations.

### 2.4. Sensory Analysis

Sensory analysis was conducted using a five-point scale rating. A total of 17 well-trained panelists, of which 9 were females and 8 were males, all staff members of Calbee, Inc., participated in the analysis. The five-point scale rating (1: very slight or very poor; 5: very strong or very good) was based on the following attributes of the cricket powder: shrimp odor, fishy smell of seafood, oxidized flavor, roasting aroma, putrid odor·sweat·natto and overall balance. The panelists held the odor bag containing the cricket powder 20 cm away from their noses and smelled it for evaluation.

### 2.5. Color Measurement

The color of the cricket powder was measured using a CR-400 Chroma Meter (Konica Minolta, Inc., Tokyo, Japan), having a color system based on the CIE *L** (lightness), *a** (red/green coordinate) and *b** (yellow/blue coordinate) color space. The colorimeter was previously calibrated using its own white calibration system. Additionally, the total color difference (Δ*E*) was calculated using the following formula:Δ*E* = (Δ*L**^2^ + Δ*a**^2^ + Δ*b**^2^)^1/2^

### 2.6. Sugar Content

An aliquot of 5 g of each sample was added to 25 mL of 40% ethanol aqueous solution (*v*/*v*). The mixture was shaken vertically for 15 min at room temperature using a RECIPRO SHAKER/SR-2s (TAITEC CORPORATION, Saitama, Japan). The supernatant was collected and filtrated through a 0.45 µm PTFE membrane filter (Merck KGaA, Darmstadt, Germany). The chromatographic separation was performed using an Agilent 1260 Infinity coupled with an Agilent 1290 Infinity Evaporative Light Scattering Detector (ELSD) (Agilent Technologies, Inc., Santa Clara, CA, USA). The system was equipped with Shodex HILICpak VG-50 4E (250 mm × 4.6 mm, 5 µm; Resonac Corporation, Tokyo, Japan) and Shodex HILICpak VG-50G 4A (10 mm × 4.6 mm; Resonac Corporation) with a flow rate of 1.2 mL/min at 40 °C. The elution was performed with a 75% acetonitrile aqueous solution for 15 min. The ELSD conditions were as follows: the evaporator temperature was 70 °C, the nebulizer temperature was 50 °C, and the gas flow rate was 1.20 SLM.

### 2.7. Total Phenolic Content

An aliquot of 2 g of each sample was homogenized for 3 min with 25 mL of *n*-hexane for defatting. After centrifugation, the supernatant was removed. The defatting process was performed three times. The residue was collected and left under a fume hood at room temperature for 1 day. The dry defatted sample was homogenized for 3 min with 25 mL of 70% methanol aqueous solution (*v*/*v*). After centrifugation, the supernatant was collected. The extraction process was performed three times. The collected solutions were combined, and the volume was fixed at 100 mL. Afterwards, 100 µL of extract or standard (gallic acid monohydrate) and 2 mL of 70% methanol aqueous solution were mixed and then added to 500 µL of 0.4 N Folin–Ciocalteu reagent and 500 µL of 10% sodium carbonate aqueous solution. After having been vortexed for a few seconds, the mixture was left to react for 1 h in the dark at room temperature. The absorbance at 760 nm was measured using a Beckman DU 800 Spectrophotometer (Beckman Coulter, Inc., Brea, CA, USA).

### 2.8. Statistical Analysis

For cricket powder data interpretation, the results are expressed as means ± standard deviations for powder from crickets reared in separate cages (*n* = 3). In order to evaluate the effects of different quantities of each cricket powder, data were analyzed using Welch’s *t*-test (statistical program R 4.1.1, R Foundation for Statistical Computing, Vienna, Austria). The results were considered statistically significant at *p* < 0.05.

## 3. Results

### 3.1. Cricket Powder Fatty Acids Content

The fatty acid content in each cricket powder analyzed is shown in Table 1. In total, 24 compounds belonging to three different categories, namely, saturated, monounsaturated and polyunsaturated fatty acids, were identified in both sample powders. The major compounds were linoleic acid (C18:2), oleic acid (C18:1), palmitic acid (C16) and stearic acid (C18), which accounted for more than 90% of the fatty acid composition. However, no differences in their contents were observed. The levels of polyunsaturated fatty acids, specifically γ-linolenic acid (C18:3), α-linolenic acid (C18:3), arachidonic acid (C20:4) and docosahexaenoic acid (C22:6), were lower in ACP than in CCP. The total fatty acid contents were similar, with only the *n*-3 polyunsaturated fatty acid level being lower and Σ *n*-6/*n*-3 extremely higher in ACP than in CCP.

### 3.2. Cricket Powder Volatile Profiles

The volatile compounds found in each cricket powder are listed in Table 2. A total of 50 volatile compounds were identified, of which 11 (2-butenal, 2-methyl-; 2-hexanone; cyclohexanone; 3-ethyl-3-methylheptane; 4,4-dimethyl octane; dodecane; dodecane, 4-methyl-; tridecane, 6-methyl-; dimethyl sulfone; (2-aziridinylethyl)amine; and pentanoic acid, 5-hydroxy-, 2,4-di-t-butylphenyl esters) were only present in CCP and another 11 (benzeneacetaldehyde; acetaldehyde; 2-butanone; ethanone, 1-(2,3-dihydro-1H-inden-5-yl)-; 2,4-dimethyl-1-heptane; octane, 4-ethyl-; octane, 4-methyl-; 2-(3-methylbutyl)-3,5-dimethylpyrazine; dimethyl sulfide; 3-carene; and 3,5-dimethyldihydropyran-2,6-dione) were only found in ACP. In both sample powders, the major groups were aldehydes, hydrocarbons and acids. In ACP, hydrocarbons, amines and ethers accounted for a lower composition rate and ketones for a higher one than in CCP. The major compounds were acetic acid, hexanal, butanal, 3-methyl and decane in both sample powders, but no differences were observed. Lower contents of 2-decanone, decane, 5-methyl-6-methylene-, undecane, 5-methyl- and furan, 2-pentyl- were found in ACP than in CCP.

### 3.3. Sensory Analysis

To characterize the smell of each cricket powder, six items (shrimp odor, fishy smell of seafood, oxidized flavor, roasting aroma, putrid odor·sweat·natto and overall balance) were evaluated using sensory analysis. Compared with CCP, ACP scored lower in the fishy smell of seafood and putrid odor·sweat·natto and higher in oxidized flavor and overall balance, as shown in Figure 1. There were no differences among samples with respect to shrimp odor and roasting aroma scores.

### 3.4. Cricket Powder Color Characteristics

The color parameters and photographs of each cricket powder are shown in Table 3 and Figure 2, respectively. A visual comparison of the color of CCP and ACP showed that ACP was darker than CCP. The Δ*E** value (total color difference), which indicates that there is a color difference if it is greater than 1, was 8.01. Compared with CCP, the *L** value (luminosity) of ACP was lower, but its *a** value (red/green coordinate) was higher.

### 3.5. Cricket Powder Sugar Content

The sugar content of each cricket powder is shown in Table 4. Glucose and trehalose were present in both cricket powders, while fructose and sucrose were only found in ACP. Maltose was not detectable in either sample. Total sugar content was higher in APC than in CCP. No significant difference was observed in the content of glucose and trehalose.

### 3.6. Cricket Powder Total Phenolic Content

The total phenolic contents of each cricket powder are shown in Table 5. No significant difference was observed for the total phenolic contents of either cricket powder.

## 4. Discussion

Fatty acids are considered substrates for SOPs, which represent one of the factors that greatly affect food flavor. Edible crickets contain about 10~30% fatty acids in dry matter equivalent, whose content and composition vary greatly depending on the diet the animals are fed during their growth phase [10,13]. In this study, the total amount of fatty acids in both cricket powders did not change, but γ-linolenic acid, α-linolenic acid, arachidonic acid and docosahexaenoic acid, which are all polyunsaturated fatty acids, were lower in the powder from apple-by-product-fed crickets (Table 1). To examine the impact of the diet fed to crickets during their growth phase on their fatty acid contents, we analyzed the fatty acid composition in the control diet (CD) and apple by-products (ABPs) used in this study (Appendix A). Most of the fatty acid contents of the ABPs were much lower than in the CD. Compared with the latter, the γ-linolenic acid content in the ABPs was significantly lower, and arachidonic acid and docosahexaenoic acid were not detected. These results are consistent with a previous report, which concluded that changing the fatty acid composition in the diet during the growth phase altered the contents in crickets [13]. This suggests that the low unsaturated fatty acid content in ABPs prevents crickets from utilizing the fatty acids available in the diet. Conversely, there is also the possibility that the high polyunsaturated fatty acid content in CCP was due to their high content in the CD. Gut loading, a method in which crickets are fed a highly nutritious diet just prior to harvest to accumulate nutrients in their gut, has also been used in previous research [21]. The higher polyunsaturated fatty acid content in CCP could have been due to the remaining CD in their gut. Polyunsaturated fatty acids with many double bonds are known to be sensitive to oxygen, producing oxidation products that contribute to bad odor and unpleasant taste [22,23]. Rearing crickets on food waste with low unsaturated fatty acid content, such as ABPs, may lower the unsaturated fatty acid content in these animals and reduce the impact of oxidation products on flavor.

Cricket volatile compounds are among the factors affecting flavor; for instance, they are responsible for cricket-related odor [24]. In previous research, approximately 40–60 volatile compounds have been identified in edible cricket powder, 14 of which have also been determined as being related to cricket-related odor [24,25,26]. In this study, we identified approximately 50 volatile compounds in both cricket powders (Table 2). Although several of the compounds detected in previous studies, such as furan, 2-pentyl- and hexanal, were also detected in this study, there were significant variations in the number of detected compounds and their composition. In addition, we were unable to detect the 14 known cricket-related odor compounds in both cricket powders. This result may be attributed to the large variation in cricket species and processing conditions.

In this study, the volatile compound composition differed between CCP and ACP (Table 2). To examine the effect of volatile compounds in the diet during the growth phase, we analyzed the volatile compounds in the CD and ABPs (Appendix A). Among those that were only found in CCP, 2-butenal, 2-methyl-, dodecane, dodecane, 4-methyl-, tridecane, 6-methyl- and (2-aziridinylethyl)amine were detected only in the CD. Acetaldehyde and dimethyl sulfide, which were exclusive to ACP, were detected only in the ABPs. However, the presence of the several compounds we found could not be explained in relation to the diets. For example, 3-carene was present in both the CD and ABPs, but it was detected only in ACP. These inconsistent results could be attributed to the following factors: (1) the detected compounds resulted from the accumulation of dietary-origin compounds; (2) the diet-derived compounds were metabolized in the gut of crickets, and the metabolites were detected as volatile compounds; (3) the compounds detected had thermally degraded during processing. However, no clear factors could be identified. In this study, since the species, growth conditions—except for the diet—and processing conditions were the same in both cricket groups, the differences in volatile compounds were certainly caused by differences in the diets fed to the crickets during their growth phase; however, the specific chemical reactions could not be ascertained because it was not possible to evaluate the changes that occurred during processing.

Flavor is mainly evaluated using sensory or instrumental analysis. In particular, sensory tests are highly reliable and allow for the identification of flavors perceived by humans [27]. In this study, sensory analysis was conducted on CCP and ACP. The results show that ACP’s shrimp odor and putrid odor·sweat·natto were reduced and its overall balance was improved compared with CCP (Figure 1). It is known that one of the reasons consumers are reluctant to consume insects is concern about their flavor [28]. The results of this study suggest that ACP has the potential to improve consumer acceptance. The differences in flavor between CCP and ACP may be attributed to the volatile compounds. For example, the following differences may have affected the flavor: ACP had a higher content of ketones and contained acetoaldehyde (a fruity and pleasant odor), 2-butanone (a sharp sweet odor), etc. On the other hand, CCP had higher contents of carbohydrates, amines and furans and contained 2-hexanone (a sharp odor), among others. However, it is generally acknowledged that it is extremely difficult to explain the relationship between the flavor perceived by humans and flavor-related volatile compounds for the following reasons: (1) there are multiple volatile compounds involved in flavor perceived by humans; (2) combinations of volatile compounds can result in different flavors from those resulting from individual compounds [27]. Therefore, we were unable to identify the volatile compounds that affected flavor in this study. However, it was found that differences in the diets administered during the growth phase affected not only the volatile compound composition but also the flavor perceivable by humans.

Previous studies have shown that increasing the proportion of cricket powder added to food products results in a darker color, which is considered to be due to caramelization or the Maillard reaction [29]. The same explanation was considered for the color difference between CCP and ACP in this study (Table 3, Figure 2). Therefore, we analyzed the sugar content, which plays an important role in these reactions (Table 4), and found that the content of the reducing sugar, i.e., fructose, differed between the sample powders, suggesting that the reactivity of caramelization or Maillard reaction may have differed. Furthermore, the oxidation of polyphenols is known to be one of the factors causing an off-color appearance [30,31]. Since ABPs contain polyphenols, the possibility of polyphenol-induced browning was considered, but no difference in the total polyphenol values between the two sample powders was observed (Table 5). This result suggests that polyphenol may not have had a great effect on the color differences observed in this study. Since it was not possible to investigate the changes that occurred during processing, we could not determine a detailed reaction, but we believe that the color change may have been caused by caramelization and the Maillard reaction due to the difference in compound contents in the diets. Food color is an important factor influencing consumer choice upon purchase [32], and the color difference between CCP and ACP may affect consumer acceptability when utilized in processed foods.

ABPs are defined as residues from the processing of apple products and represent a potential source of bioactive compounds [33,34,35]. The ABPs used in this study were food-processing waste materials, which are usually discarded. In this study, we not only characterized powder from crickets reared on ABPs but also revealed the potential value of these by-products as components for cricket diets.

## 5. Conclusions

In this study, we characterized ACP and CCP in terms of flavor and appearance. The flavor and appearance of both cricket powders were significantly different, which seemed to be due to the diet crickets were fed during their growth phase. However, detailed effects could not be determined because we were unable to evaluate the changes that occurred during processing. Thus, further research is needed.

This study indicated that, in addition to cricket species and processing conditions, the choice of the diet used to rear edible crickets is also an important factor.

## Figures and Tables

**Figure 1 foods-13-01668-f001:**
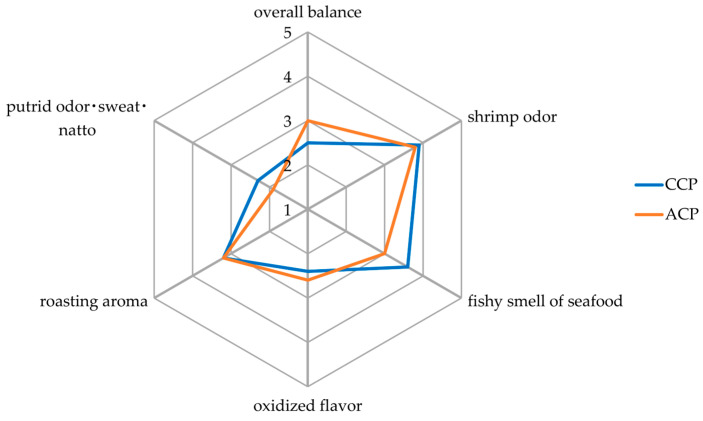
Organoleptic characteristics of cricket powder. CCP: control cricket powder; ACP: apple cricket powder; 1: very slight or very poor; 5: very strong or very good.

**Figure 2 foods-13-01668-f002:**
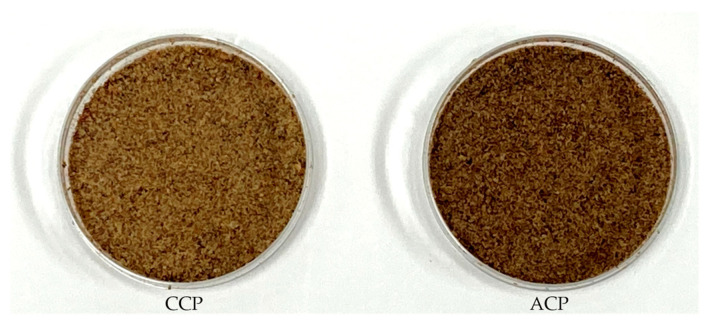
Cricket powders. CCP: control cricket powder; ACP: apple cricket powder.

**Table 1 foods-13-01668-t001:** Fatty acid contents (mg/kg DM).

Shorthand Nomenclature	Fatty Acid Name	Type	CCP	ACP
12:0	Lauric acid	SFA	171.3 ± 14.1	226.9 ± 33.6
14:0	Myristic acid	SFA	290.1 ± 27.5	299.1 ± 41.5
14:1 (*n*-5)	Myristoleic acid	MUFA	23.7 ± 3.6	40.4 ± 18.0
15:0	Pentadecanoic acid	SFA	39.1 ± 5.0	32.9 ± 2.7
15:1 (*n*-5)	Pentadecenoic acid	MUFA	13.8 ± 1.5	13.0 ± 1.4
16:0	Palmitic acid	SFA	11,247.4 ± 1172.4	12,356.6 ± 811.7
16:1 (*n*-7)	Palmitoleic acid	MUFA	499.5 ± 30.5	543.2 ± 23.3
17:0	Margaric acid	SFA	119.6 ± 11.0	118.8 ± 16.2
18:0	Stearic acid	SFA	4685.4 ± 324.3	4676.8 ± 238.6
18:1 (*n*-9)	Oleic acid	MUFA	11,711.0 ± 1007.3	12,262.9 ± 1146.4
18:2 (*n*-6)	Linoleic acid	PUFA	18,886.9 ± 1432.0	20,999.5 ± 1091.7
18:3 (*n*-6)	γ-Linolenic acid	PUFA	443.9 ± 88.4	204.3 ± 28.9 *
18:3 (*n*-3)	α-Linolenic acid	PUFA	53.4 ± 7.1	34.5 ± 2.0 *
20:1 (*n*-9)	Eicosenoic acid	MUFA	80.9 ± 8.6	95.6 ± 11.5
20:2 (*n*-6)	Eicosadienoic acid	PUFA	106.5 ± 14.2	141.7 ± 17.8
20:3 (*n*-6)	Eicosatrienoic acid	PUFA	144.5 ± 8.5	142.5 ± 12.3
20:3 (*n*-3)	Eicosatrienoic acid	PUFA	10.4 ± 1.9	9.1 ± 1.8
20:4 (*n*-6)	Arachidonic acid	PUFA	595.5 ± 60.8	448.2 ± 19.4 *
20:5 (*n*-3)	Eicosapentaenoic acid	PUFA	12.4 ± 1.6	11.2 ± 0.6
21:0	Heneicosanoic acid	SFA	37.4 ± 11.2	33.0 ± 2.7
22:0	Behenic acid	SFA	17.4 ± 2.1	16.5 ± 3.4
22:1 (*n*-9)	Erucic acid	MUFA	8.8 ± 1.5	4.9 ± 8.4
22:6 (*n*-3)	Docosahexaenoic acid	PUFA	78.1 ± 7.2	15.6 ± 5.1 *
24:1 (*n*-9)	Nervonic acid	MUFA	37.6 ± 4.0	33.5 ± 2.1
Total			49,314.6 ± 4150.2	52,760.6 ± 3404.7
Σ SFAs			16,607.6 ± 1527.5	17,760.5 ± 1121.6
Σ MUFAs			12,375.3 ± 1041.5	12,993.5 ± 1158.1
Σ PUFAs			20,331.7 ± 1593.7	22,006.6 ± 1151.7
Σ *n*-3 PUFAs			154.3 ± 15.0	70.4 ± 9.0 *
Σ *n*-6 PUFAs			20,177.4 ± 1579.4	21,936.2 ± 1142.7
Σ *n*-6/*n*-3			130.9 ± 4.2	313.9 ± 25.2 *

Values are expressed as means ± standard deviations for powder from crickets reared in separate cages (*n* = 3). Powder from crickets from the same cage was measured in triplicate; * indicates significant differences between powders (*p* < 0.05). CCP: control cricket powder; ACP: apple cricket powder; SFAs: saturated fatty acids; MUFAs: monounsaturated fatty acids; PUFAs: polyunsaturated fatty acids; DM: dry matter.

**Table 2 foods-13-01668-t002:** Volatile profiles of cricket powders.

Volatile Compounds	CCP	ACP
Aldehydes		
Propanal, 2-methyl- ^a^	3.83 ± 0.45	5.15 ± 0.71
Benzeneacetaldehyde ^a^	n.d.	0.01 ± 0.00
2-Butenal, 2-methyl-	0.10 ± 0.02	n.d.
Acetaldehyde	n.d.	0.83 ± 0.12
Butanal, 2-methyl- ^a^	6.82 ± 1.22	8.33 ± 1.40
Butanal, 3-methyl- ^a^	10.22 ± 1.46	11.34 ± 1.12
Heptanal	0.33 ± 0.07	0.45 ± 0.04
Hexanal ^b^	11.24 ± 2.79	14.97 ± 3.44
Nonanal ^b^	0.42 ± 0.04	0.43 ± 0.04
Pentanal ^b^	2.22 ± 0.43	2.78 ± 0.12
Total	35.17 ± 4.50	44.30 ± 1.04
Ketones		
2-Butanone	n.d.	3.27 ± 0.33
2-Decanone	0.28 ± 0.05	0.18 ± 0.02 *
2-Heptanone	2.47 ± 0.90	2.06 ± 0.17
2-Hexanone	0.15 ± 0.02	n.d.
Acetone	3.29 ± 0.85	3.89 ± 0.64
Cyclohexanone	0.14 ± 0.04	n.d.
Ethanone, 1-(2,3-dihydro-1H-inden-5-yl)-	n.d.	0.02 ± 0.01
Total	6.32 ± 0.81	9.42 ± 1.08 *
Hydrocarbons		
3-Ethyl-3-methylheptane	0.03 ± 0.01	n.d.
4,4-Dimethyl octane	0.26 ± 0.14	n.d.
5-Ethyldecane	0.19 ± 0.01	0.16 ± 0.03
Decane	11.56 ± 1.89	10.34 ± 0.90
Decane, 5-methyl-6-methylene-	0.18 ± 0.01	0.12 ± 0.00 *
Dodecane	5.28 ± 0.24	n.d.
Dodecane, 2,7,10-trimethyl-	3.74 ± 0.78	3.14 ± 0.56
Dodecane, 4,6-dimethyl-	0.31 ± 0.14	0.17 ± 0.02
Dodecane, 4-methyl-	0.06 ± 0.03	n.d.
Heptane, 2,2,4,6,6-pentamethyl-	0.73 ± 0.21	0.47 ± 0.06
Heptane, 2,4-dimethyl-	0.31 ± 0.15	0.32 ± 0.02
Nonane, 2-methyl-	0.19 ± 0.10	0.20 ± 0.04
Octane, 3,5-dimethyl-	0.17 ± 0.11	0.17 ± 0.04
Tetradecane	0.06 ± 0.10	0.00 ± 0.00
Tridecane, 6-methyl-	0.11 ± 0.01	n.d.
Undecane, 5,7-dimethyl-	0.22 ± 0.09	0.17 ± 0.03
Undecane, 5-methyl-	0.27 ± 0.01	0.20 ± 0.02 *
2,4-Dimethyl-1-heptene	n.d.	0.06 ± 0.01
Octane, 4-ethyl-	n.d.	0.20 ± 0.03
Octane, 4-methyl-	n.d.	0.09 ± 0.01
Total	23.67 ± 2.67	15.81 ± 1.71 *
Benzenoids		
Benzene, 1,2,4-trimethyl-	0.08 ± 0.02	0.07 ± 0.01
Benzene, 1,3-bis(1-methylethenyl)-	0.02 ± 0.01	0.02 ± 0.00
Benzene, 1,3-dimethyl-	0.23 ± 0.10	0.13 ± 0.02
Benzene, 1,4-diethyl-	0.05 ± 0.01	0.04 ± 0.01
Benzene, 1-ethynyl-4-methyl-	0.04 ± 0.01	0.04 ± 0.00
Ethylbenzene	0.37 ± 0.22	0.19 ± 0.05
Total	0.80 ± 0.36	0.48 ± 0.09
Pyrazines		
Pyrazine ^a^	0.12 ± 0.02	0.12 ± 0.02
Pyrazine, 2,5-dimethyl- ^a^	0.68 ± 0.22	0.60 ± 0.08
Pyrazine, 3-ethyl-2,5-dimethyl- ^a^	0.26 ± 0.03	0.29 ± 0.02
Pyrazine, methyl- ^a^	0.71 ± 0.27	0.56 ± 0.06
2-(3-Methylbutyl)-3,5-dimethylpyrazine ^a^	n.d.	0.03 ± 0.01
Total	1.77 ± 0.53	1.59 ± 0.18
Sulfur compounds		
Dimethyl sulfide	n.d.	0.30 ± 0.16
Dimethyl sulfone	0.18 ± 0.04	n.d.
Disulfide, dimethyl	0.59 ± 0.19	0.29 ± 0.11
Total	0.77 ± 0.21	0.60 ± 0.15
Acids		
Acetic acid	22.14 ± 6.97	22.87 ± 1.80
Butanoic acid, 3-methyl-	0.10 ± 0.01	0.10 ± 0.03
Total	22.24 ± 6.97	22.97 ± 1.82
Amines		
(2-Aziridinylethyl)amine	3.23 ± 1.13	n.d.
Methylamine, N,N-dimethyl-	1.95 ± 0.14	1.65 ± 0.42
Total	5.18 ± 1.10	1.65 ± 0.42 *
Terpenes		
D-Limonene	0.16 ± 0.03	0.12 ± 0.02
3-Carene	n.d.	0.09 ± 0.01
Total	0.16 ± 0.03	0.21 ± 0.02
Ethers		
3,5-Dimethyldihydropyran-2,6-dione	n.d.	0.19 ± 0.02
Furan, 2-pentyl- ^a^	3.33 ± 0.54	1.91 ± 0.23 *
Total	3.33 ± 0.54	2.10 ± 0.24 *
Esters		
Pentanoic acid, 5-hydroxy-, 2,4-di-t-butylphenyl esters	0.03 ± 0.01	n.d.
Total	0.03 ± 0.01	n.d.
Alcohols		
1-Pentanol	0.56 ± 0.06	0.88 ± 0.30
Total	0.56 ± 0.06	0.88 ± 0.30

Values are expressed as means ± standard deviations for powder from crickets reared in separate cages (*n* = 3). Powder from crickets from the same cage was measured in quintuplicate; * indicates significant difference between powders (*p* < 0.05); ^a^ indicates Maillard reaction and Strecker degradation products; ^b^ indicates secondary oxidation products. CCP: control cricket powder; ACP: apple cricket powder; n.d.: not detected.

**Table 3 foods-13-01668-t003:** The results of the color measurement.

Sample Powders	Color Parameters			
	*L**	*a**	*b**	Δ*E**
CCP	42.23 ± 0.87	5.57 ± 0.55	20.2 ± 0.87	-
ACP	34.40 ± 0.79 *	6.90 ± 0.22 *	19.21 ± 0.15	8.01

Values are expressed as means ± standard deviations for powder from crickets reared in separate cages (*n* = 3). Powder from crickets from the same cage was measured in triplicate; * indicates significant difference between powders (*p* < 0.05). CCP: control cricket powder; ACP: apple cricket powder; *L**: luminosity; *a**: red/green coordinate; *b**: yellow/blue coordinate; Δ*E**: total color difference.

**Table 4 foods-13-01668-t004:** Sugar contents (g/100 g DM).

Sugar	CCP	ACP
Glucose ^a^	0.42 ± 0.02	0.37 ± 0.06
Fructose ^a^	n.d.	0.39 ± 0.11
Sucrose	n.d.	0.25 ± 0.04
Maltose ^a^	n.d.	n.d.
Trehalose	0.18 ± 0.04	0.20 ± 0.05
Total	0.70 ± 0.05	1.22 ± 0.20 *

Values are expressed as means ± standard deviations for powder from crickets reared in separate cages (*n* = 3). Powder from crickets from the same cage was measured in triplicate; * indicates significant differences between powders (*p* < 0.05); ^a^ indicates reducing sugar. CCP: control cricket powder; ACP: apple cricket powder; n.d.: not detected.

**Table 5 foods-13-01668-t005:** Total phenolic contents (g GAE/100 g DM).

Samples	CCP	ACP
Total phenolic contents	0.42 ± 0.02	0.40 ± 0.01

Values are expressed as means ± standard deviations for powder from crickets reared in separate cages (*n* = 3). Powder from crickets from the same cage was measured in triplicate. GAE: gallic acid equivalents; CCP: control cricket powder; ACP: apple cricket powder; DM: dry matter.

## Data Availability

The data presented in this study are available on request from the corresponding author. The data are not publicly available due to privacy restrictions.

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
