# Peer review of "Analysis of Fatty Acid Composition and Volatile Profile of Powder from Edible Crickets (Acheta domesticus) Reared on Apple By-Products"

_foods, 2024, doi:10.3390/foods13111668_

Round 1

Reviewer 1 Report

Comments and Suggestions for Authors

The manuscript is focused on nutritional values and flavour of edible crickets reared on apple by-products. The text is written at a good level. The chosen analytical methods are adequate. The results are appropriately presented, properly evaluated, discussed, and commented on. The conclusions are succinctly summarized. The comments below:

L 73: Replace “growing environments” by “rearing conditions”.

L 89: Were the insects starved before being harvested? For how long?

L 89 and 92: Please specify the used freezers and other equipment.

L 92: For how long were the samples frozen?

L 92: You had three biological replicates. How many analytical replicates did you do?

L 98: Each used chemical must be specified (purity, producer).

L 142: How were the descriptors selected for sensory evaluation?

L 162: Extraction or chromatographic separation?

L 180: Use molarity instead of normality.

L 205: The values in the table are very low. Perhaps it would be better to express them in milligrams per kilogram.

L 228: Are there any units to the presented values?

L 282: Insects are usually starved before being harvested. Did you do it in your experiment?

Author Response

Thank you for reading our manuscript and for giving us accurate and polite advice.

We also appreciate the time and effort you have dedicated to providing insightful feedback on ways to strengthen our manuscript.

Thus, it is with great pleasure that we resubmit our article for further consideration.

We have incorporated changes that reflect the detailed suggestions you have graciously provided.

We also hope that our edits and the responses we provide below satisfactorily address all the issues and concerns you have noted. I could hope you can check it in the re-submitted files

Reviewer 2 Report

Comments and Suggestions for Authors

Abstract need to improve in terms of information of  Edible Crickets.

line #25 what types of chemical composition & why do you think ?

line #37 reference ?

line #48 what is the percentage of amino acids, minerals and fatty acids ? 

#Material section need to improve.

#Results section, the title is totally incorrect. Every title should be one theme of result.

line #235 do you have any reference ?

Figure 2 look like no significant difference ? why ?

Conclusion section need to organise. 

You used 50 volatile compounds. I think it is too low. Could you please increase the amount around 100. 

Comments on the Quality of English Language

ok

Author Response

(The authors gave the same response as above.)

Round 2

Reviewer 2 Report

Comments and Suggestions for Authors

Many things already improved. But still I am worry about the title of every result. However good scientific research.